# TreeDQN: Learning to minimize Branch-and-Bound tree

## Abstract

Combinatorial optimization problems require an exhaustive search to find the optimal solution. A convenient approach to solving combinatorial optimization tasks in the form of Mixed Integer Linear Programs is *Branch-and-Bound*. Branch-and-Bound solvers split a task into two parts by dividing the domain of an integer variable, and solve them recursively, producing a tree of nested sub-tasks. The efficiency of the solver depends on the *branching heuristic* used to select a variable for splitting. In the present work, we propose TreeDQN - a reinforcement learning method that can efficiently learn the branching heuristic. We view the variable selection task as a tree Markov Decision Process, prove the contraction property of the Bellman operator in the tree Markov Decision Process, and propose a modified learning objective for the reinforcement learning agent. Our agent requires less training data and produces smaller trees compared to previous reinforcement learning methods.

## 1 Introduction

Practical applications in multiple areas such as logistics (Bertsimas & Van Ryzin, 1991), portfolio management (Markowitz, 1952), manufacturing (Barahona et al., 1988), and others share a combinatorial structure. Finding the optimal solution for a combinatorial task requires an exhaustive search over all valid combinations of variables. The optimal solution for a combinatorial problem formulated as a Mixed Integer Linear Program (Wolsey & Nemhauser, 1999) can be efficiently obtained with the *Branch-and-Bound* algorithm (B&B) (Land & Doig, 2010). The B&B algorithm employs *divide-and-conquer* approach. At each step, it splits the domain of one of the integer variables and eliminates paths that can not lead to a feasible solution. The performance of the B&B algorithm depends on two sequential decision-making processes: variable selection and node selection. Node selection picks the next node in the B&B tree to evaluate, and variable selection chooses the next variable to split on (Linderoth & Savelsbergh, 1999). The optimal variable selection method, frequently dubbed as *branching rule*, will lead to smaller trees and a more efficient solver. Although the optimal *branching rule* is not known (Lodi & Zarpellon, 2017), all modern solvers implement human-crafted heuristics, which were designed to perform well on a wide range of tasks (Gleixner et al., 2021). At the same time, practitioners frequently solve the same task with different parameters, so the branching rule adapted to a specific distribution of tasks may lead to a significant performance boost and business impact. The branching rule is applied sequentially to minimize the resulting tree size, which resembles the reinforcement learning paradigm in which an agent interacts with the environment to maximize the expected return. Recently reinforcement learning achieved state-of-the-art results in a diverse set of tasks, from beating world champions in the games of Go (Silver et al., 2018) and Dota2 (Berner et al., 2019) to aligning optical interferometer (Sorokin et al., 2020), controlling nuclear fusion reactor (Degrave et al., 2022), tuning hyperparameters of simulated quantum annealers (Beloborodov et al., 2021) and optimizing output of large language models (OpenAI, 2023).

However, the direct application of reinforcement learning methods to variable selection is challenging. The main difficulties are the tree structure of the B&B algorithm, the high variance of the distribution of resulting tree sizes, and the computational complexity of the B&B algorithm. In our work, we view the variable selection task as a *tree Markov Decision Process* (tree MDP). In the tree MDP, instead of a single next state the agent receives multiple next states — descendants of the current tree node. We prove the contraction property of the Bellman operator in the tree MDP and develop a

sample efficient off-policy reinforcement learning method. To overcome the high variance of tree size distribution we propose a modified loss function that optimizes the geometric mean of expected return. As a result, our method trains more stable, produces smaller trees, and is more sample efficient than the previous RL methods.

## 2 BACKGROUND

### 2.1 MIXED INTEGER LINEAR PROGRAMMING

A *Mixed Integer Linear Program* (MILP) is a non-convex optimization problem of the form:

$$\min\Big\{c^\top x \colon Ax \le b\,, x \in \big[l, u\big]\,, x \in \mathbb{Z}^m \times \mathbb{R}^{n-m}\Big\}, \tag{1}$$

where $c \in \mathbb{R}^n$ is the objective coefficient vector, $b \in \mathbb{R}^m$ is the right-hand-side constraint, $A \in \mathbb{R}^{m \times n}$ is the constraint matrix, $l, u \in \mathbb{R}^n$ are the lower and upper bound vectors and $m \ge 1$ is the number of components of vector $x$ restricted to be integer. The method of choice to find the optimal solution of a MILP is Branch-and-Bound (Land & Doig, 2010). B&B builds a tree of nested MILP subproblems with non-overlapping feasibility sets. For each subproblem, the algorithm computes a lower (dual) bound and a global upper (primal) bound. The lower bound (LB) is obtained as an optimal solution for LP relaxation of the subproblem. LP relaxation considers all discrete variables as continuous and maintains all the other constraints. The global upper bound (GUB) is obtained as a minimum of found feasible solutions. A solution of LP relaxation is considered feasible if it satisfies the integrality constraints of the original problem. B&B uses these bounds to enforce efficiency by pruning the tree. Pruning discards subtrees that can not contain a feasible solution better than the current GUB. Visiting every open node guarantees that the B&B eventually finds the best integer-feasible solution. The algorithm can be described in the following steps:

1. The root of the tree is the original MILP problem. GUB is set to $+\infty$.
2. Select not visited node from the tree using *node selection strategy*. Compute LB as a solution of the relaxed problem: $\min(c^T \hat{x} : A\hat{x} \le b, \hat{x} \in [l, u], \hat{x} \in \mathbb{R}^n)$.
3. Update GUB if the relaxed problem provides a feasible solution.
4. If LB < GUB and corresponding MILP is feasible, choose one of the fractional variables $\hat{x}_i$ using the *branching rule* and split its domain into two parts, which produces two nodes with constraints $l_i \le x_i \le \lfloor \hat{x}_i \rfloor$ and $\lceil \hat{x}_i \rceil \le x_i \le u$. Add descendant nodes to the tree.
5. Mark the node as visited.
6. Go back to the 2nd step.

The resulting efficiency of the algorithm depends on the *node selection strategy*, which arranges the open leaves for visiting, and the *branching rule*, which selects an integer variable for splitting. Practical implementations of the Branch-and-Bound algorithm in SCIP (Bestuzheva et al., 2021) and CPLEX (Cplex, 2009) solvers rely on heuristics for node selection and variable selection. A straight forward strategy for node selection is *Depth-First-Search* (DFS), which aims to find any integer feasible solution faster to prune branches that do not contain a better solution. In the SCIP solver, the default node selection heuristic tries to estimate the node with the lowest feasible solution. One of the best-known general heuristics for the variable selection is *Strong Branching*. It is a tree-size efficient and computationally expensive branching rule (Achterberg, 2007). For each fractional variable with integrality constraint, Strong Branching computes the lower bounds for the left and right child nodes and uses them to choose the variable for splitting.

### 2.2 TREE MDP

The tree Markov Decision Process was proposed by Scavuzzo et al. (2022). It is defined by $(\mathcal{S}, \mathcal{A}, p_{\text{init}}, p^+, p^-, r)$, where states $s \in \mathcal{S}$, actions $a \in \mathcal{A}$, initial state distribution $p_{\text{init}}(s_0)$, probabilities of having left and right descendant nodes $p^+(s_{t+1}^+|s_t, a_t)$, $p^-(s_{t+1}^-|s_t, a_t)$, and reward function $r : S \to R$. The key difference between a temporal MDP and a tree MDP is that trajectories in the

tree MDP have a tree structure. In the tree MDP we can define value function $V(s)$ as a sum of reward and value functions of the next states:

$$V(s_t) = r(s_t, a, s_{t+1}^{\pm}) + p^+ V(s_{t+1}^+) + p^- V(s_{t+1}^-) \tag{2}$$

The goal of the agent is to find a policy that would maximize the expected return. For instance, if the reward equals $-1$, the value function equals the size of the tree with a negative sign. Hence, the agent maximizing the expected return would minimize the expected tree size.

The *variable selection* process employed by the Branch-and-Bound algorithm can be considered a tree Markov Decision Process. The *state* of the tree MDP is $(\text{MILP}_t, \text{GUB}_t)$, and the *action* is the fractional variable chosen for splitting. To guarantee the Markov property, we need probabilities $p^+$, $p^-$ that depend only on the parent state and action. In fact, they depend on the GUB, which can vary for different visiting orders. To enforce the Markov property, one can either set the GUB in the root node equal to the optimal solution or choose *Depth First Search* as the node selection strategy (Scavuzzo et al., 2022). During testing, the optimal solution for the task at hand is unknown and can not be used to set the global upper bound, which leads to a gap between training and testing environments. More efficient heuristics for *node selection* also induce a gap for an agent trained with DFS node selection strategy. This gap is often considered moderate and does not affect the performance significantly.

## 3 RELATED WORK

For the first time statistical approach to learning a branching rule was applied by Khalil et al. (2016). Authors used SVM (Cortes & Vapnik, 1995) to predict the variable ranking of an expert for a single task instance. Later works (Khalil et al., 2017) and (Selsam et al., 2018) proposed methods based on Graph Convolutional Networks (GCNN) (Kipf & Welling, 2017) to find an approximate solution of combinatorial tasks. Gasse et al. (2019) applied the same neural network architecture to imitate the Strong Branching heuristic in sophisticated SCIP solver (Bestuzheva et al., 2021). The imitation learning agent can not produce trees shorter than the expert, however, it solves the variable selection task much faster, especially if running on GPU, thereby speeding up the whole B&B algorithm significantly. Gupta et al. (2020) investigated the choice of the model architecture and proposed a hybrid model that combines the expressive power of GCNN with the computational efficiency of multi-layer perceptrons. Despite the time performance increase, imitation learning agents can not lead to better heuristics.

A more promising direction is to learn a variable selection rule for the Branch-and-Bound algorithm with reinforcement learning. In this approach, we will keep the guarantees of the B&B method to find an optimal solution and possibly speed up the algorithm significantly by optimal choices of branching variables. A natural minimization target for an agent in the B&B algorithm is the size of the resulting tree. One of the main challenges here is to map the variable selection process to the MDP and preserve the Markov property. In the B&B search trees, the local decisions impact previously opened leaves via fathoming due to global upper bound pruning which violates the Markov property. Etheve et al. (2020) proposed Fitting for Minimizing the SubTree Size (FMCTS) algorithm to learn a branching rule. In their method an agent plays an episode until termination and fits the Q-function to the bootstrapped return. Authors used the DFS node selection strategy to enforce MDP property during training. This method is sample efficient since training data can be sampled from a buffer of past experiences, but is biased because the data was obtained by older and less efficient versions of the Q-function. Following this idea Scavuzzo et al. (2022) introduced the tree MDP framework and proposed setting the global upper bound to the optimal solution for a MILP as an alternative method to enforce the MDP property. They derived policy gradient theorem for the tree MDP and evaluated REINFORCE-based agent on a set of challenging tasks similar to (Gasse et al., 2019). This method is sample inefficient since a single gradient step of the REINFORCE agent requires solving a batch of MILP tasks, but it is unbiased because agent uses only the latest data. In both works (Etheve et al., 2020) and (Scavuzzo et al., 2022) one needs a cumulative return to update the agent.

## 4 TREE DQN

The variable selection task is significantly different from an ordinary reinforcement learning environment. Hence, the successful reinforcement learning method should have the following properties:

1. Finding the optimal solution of a MILP is computationally demanding, which requires development of sample efficient reinforcement learning methods.

2. To solve an MILP task B&B method generates a tree of nested subtasks. Thus a reinforcement learning method which learns optimal branching decisions needs to map this tree structure to an episode. A direct approach to such mapping is to consider the decision-making process a tree MDP instead of a temporal MDP.

3. If a branching rule makes wrong branching decisions, it leads to weakly pruned trees with a size growing exponentially as a function of the number of integer-valued variables. So, the distribution of tree sizes produced by the B&B method with a non-perfect branching rule should have a long tail, as shown in Fig. 1. Previous works (Gasse et al., 2019; Scavuzzo et al., 2022) used the geometric mean of the final tree size to benchmark the average performance of different branching rules. This metric is more stable in the case of long-tailed distributions than the arithmetic mean. Thus, the successful reinforcement learning method should optimize the geometric mean of expected return.

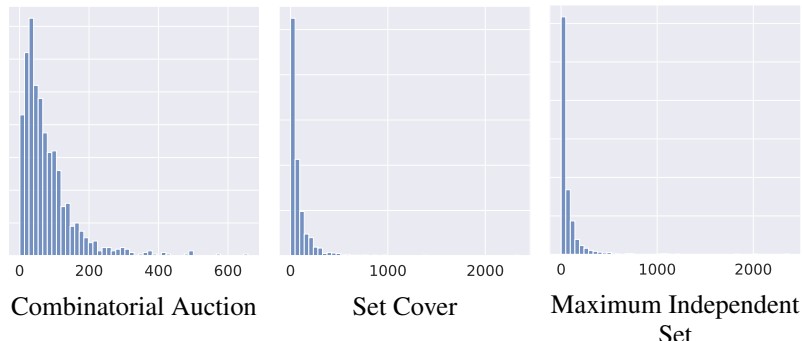

Figure 1: Distributions of tree sizes for Combinatorial Auction (Leyton-Brown et al., 2000), Set Cover (Balas & Ho, 1980), and Maximum Independent Set (Bergman et al., 2016) tasks, using Strong Branching heuristic for variable selection.

In the next sections we present our method which satisfies the requirements listed above.

### 4.1 CONTRACTION IN MEAN

From the theoretical point of view, reinforcement learning methods converge to an optimal policy due to the contraction property of the Bellman operator (Jaakkola et al., 1993). To apply RL methods to the tree MDP, we need to justify the contraction property of the tree Bellman operator.

**Theorem 4.1** Tree Bellman operator is contracting in mean.

Bellman operator for a tree MDP is defined similarly to a temporal MDP:

$$T(V(s)) = r(s, \pi(s)) + \gamma \left[ p^+ V(s^+) + p^- V(s^-) \right] \tag{3}$$

Contraction in mean is discussed, for example, in (Borovkov, 2013). Here, we will consider operator $T$ is contracting in mean if:

$$\begin{aligned} \|TV - TU\|_\infty &= p \cdot \|V - U\|_\infty, \\ \mathbb{E}\, p &< 1, \end{aligned} \tag{4}$$

where the infinity norm is defined by:

$$\|V - U\|_\infty = \max_{s \in \mathbb{S}} |V(s) - U(s)| \tag{5}$$

We will assume that the probability of having a left ($p^+$) and a right ($p^-$) child does not depend on the state. This assumption is close to the B&B tree pruning process, where the pruning decision depends on the global upper bound. Using the definition of tree Bellman operator Equation 3 and the definition of the infinity norm Equation 5 we derive the following inequality:

$$
\begin{aligned}
\|TV - TU\|_\infty &= \gamma \|p^+ V(s^+) + p^- V(s^-) - p^+ U(s^+) - p^- U(s^-)\|_\infty = \\
&\gamma \max_{s \in \mathbb{S}} \left[ p^+ |V(s^+) - U(s^+)| + p^- |V(s^-) - U(s^-)| \right] \leq \\
&\gamma (p^+ + p^-) \max_{x \in \mathbb{S}} |V(x) - U(x)|
\end{aligned}
$$

In a finite rooted tree every node except the root has exactly one incoming edge. Hence, the number of edges is one less than the number of nodes. So the expected number of child nodes $\mathbb{E}(p^+ + p^-) = \frac{N-1}{N} < 1$. This leads to the following equations:

$$
\begin{aligned}
\|TV - TU\|_\infty &= (p^+ + p^-) \cdot \|V - U\|_\infty, \\
\mathbb{E}(p^+ + p^-) &< 1,
\end{aligned}
$$

which satisfies the definition of contraction in mean (Eq. 4).

## 4.2 Loss function

Reinforcement learning methods generally regress the expected return with the mean squared error (MSE) loss function, thereby optimizing the prediction of the arithmetic mean. In the case of vast return distributions, we propose to use mean squared logarithmic error (MSLE) instead. For a variable $y$ and targets $t_i$ loss $L(y, t)$ is defined as follows:

$$L(y, t) = \text{MSE}(\log(|y|), \log(|t|)) = \frac{1}{N} \sum_i (\log(|y|) - \log(|t_i|))^2 \tag{6}$$

Since $\log(|y|) = 1/N \sum \log(|t_i|)$ minimizes the MSE function, then the optimal value for $y$ equals to geometric mean $|y| = \exp(1/N \sum_{i=1}^{N} \log(|t_i|))$. Thus, the agent trained with loss Equation 6 will be optimized to predict the geometric mean of the expected return.

In our experiments, we use a Graph Convolutional Neural Network with activation $f = -\exp(\cdot)$ applied to the output layer, which allows our agent to approximate a wide range of Q-values. For this activation function, we can implement the loss function Equation 6 numerically stable using logits before activation. Hence, the proposed loss function serves two purposes simultaneously: it optimizes the target value — geometric mean of expected return and stabilizes the learning process.

## 4.3 Implementation

In our method, we adapt the Double Dueling DQN (Mnih et al., 2015) algorithm for a tree MDP process equation 2. According to *Theorem 4.1*, the Bellman operator for a tree MDP process is contracting in mean. Hence, we can use DQN to minimize tree difference error instead of temporal difference. We significantly improve the sample efficiency by sampling previous observations from experience replay in contrast to the on-policy method proposed for variable selection (Scavuzzo et al., 2022). To stabilize the learning process in the presence of high variance returns we use MLSE loss function (Eq. 6). The whole algorithm of our method is shown in Appendix A.3, Alg. 1.

### 4.4 TRAINING

In the present work, we use an open-source implementation of the Branch-and-Bound algorithm in SCIP solver version 8.0.1 with Ecole (Prouvost et al., 2020) 0.8.1 package, which provides an interface for learning a variable selection policy. We train our agent on a set of NP-hard tasks, namely Combinatorial Auction (Leyton-Brown et al., 2000), Set Cover (Balas & Ho, 1980), Maximum Independent Set (Bergman et al., 2016), Facility Location (Cornuejols et al., 1991) and Multiple Knapsack (Fukunaga, 2009). To test the generalization ability of our agent, we evaluate the trained agent twice: (1) on the test instances from the training distribution and (2) on the large instances from the transfer distribution. Additionally, in the Appendix D, we present evaluation results of our method on a more challenging Balanced Item Placement task from ML4CO competition (Gasse et al., 2022). We use DFS for node selection during training and switch to SCIP default node selection policy for testing. Tab. 1 shows the parameters of the test and transfer distributions.

Table 1: Parameters used to generate train, test, and transfer tasks.

|  | Comb.Auct. items / bids | Set Cover rows / cols | Max.Ind.Set nodes | Facility Loc. cust. / facil. | Mult.Knap. items / knapsacks |
|---|---|---|---|---|---|
| train / test | 100 / 500 | 400 / 750 | 500 | 35 / 35 | 100 / 6 |
| transfer | 200 / 1000 | 500 / 1000 | 1000 | 60 / 35 | 100 /12 |

We use the same set of hyperparameters (see Appendix A.3 Tab. 7) to train our agent for each task distribution. Total training time did not exceed 4 days on an Intel Xeon 6326, NVIDIA A100 machine. To select the best checkpoint for testing, we perform validation using 20 fixed task instances with 5 random seeds every 50 training episodes. The validation plot in Fig. 2 shows the geometric mean of tree sizes as a function of the number of training episodes. We see that during training TreeDQN agent learns to solve variable selection tasks better, generating smaller B&B trees. The number of episodes it took to reach the best checkpoint for the agents is shown in Tab. 2.

Table 2: Number of training episodes required to reach the best checkpoint.

| Model | Comb.Auct | Set Cover | Max.Ind.Set | Facility Loc. | Mult.Knap. |
|---|---|---|---|---|---|
| TreeDQN | 700 | 800 | 800 | 200 | 850 |
| FMCTS | 1000 | 950 | 50 | 200 | 400 |
| tmdp+DFS | 22500 | 3000 | 3500 | 6500 | 9500 |

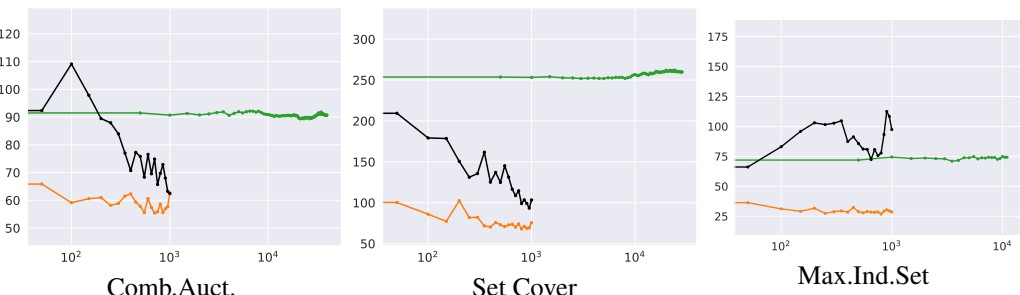

| Comb.Auct. | Set Cover | Max.Ind.Set |

Figure 2: The geometric mean of tree size as a function of a number of training episodes. Orange - TreeDQN, black - FMCTS, green - tmdp+DFS.

## 5 EVALUATION

In both test and transfer settings, we generate 40 task instances and evaluate our agent with five random seeds. We compare the performance of our TreeDQN agent with the Strong Branching rule, Imitation Learning (IL, (Gasse et al., 2019)), tMDP+DFS (on-policy, REINFORCE-based method, (Scavuzzo et al., 2022)), and FMCTS (off-policy method, (Etheve et al., 2020)) agents. For the comparison of loss functions used in TreeDQN, tMDP+DFS, and FMCTS refer to Appendix A.2.

Additionally, we provide evaluation results for the SCIP solver with default parameters. However, it is not a direct competitor to our method since internal branching rules can make several modifications to the state of the solver except branching (Gamrath & Schubert, 2018).

We present evaluation results for test instances in Tab. 3, 4. Bold numbers indicate the best-performing reinforcement learning method (TreeDQN, FMCTS of tmdp+DFS). In the Tab. 3, we show the geometric mean of the final tree size and geometric standard deviation. Tab. 3 shows that the TreeDQN agent significantly exceeds the results of the tmdp+DFS and FMCTS agents in all test tasks. The TreeDQN agent is close to the Imitation Learning agent in the first four tasks and substantially outperforms the Imitation Learning and Strong Branching in the Multiple Knapsack task.

Table 3: Geometric mean with geometric std for test tasks.

| Model | Comb.Auct | Set Cover | Max.Ind.Set | Facility Loc. | Mult.Knap. |
|---|---|---|---|---|---|
| SCIP default | $6 \pm 4$ | $12 \pm 4$ | $16 \pm 6$ | $53 \pm 12$ | $234 \pm 5$ |
| Strong Branching | $48 \pm 3$ | $43 \pm 2$ | $40 \pm 4$ | $294 \pm 9$ | $700 \pm 10$ |
| IL | $56 \pm 3$ | $53 \pm 2$ | $42 \pm 5$ | $323 \pm 8$ | $670 \pm 9$ |
| TreeDQN | $\mathbf{58 \pm 3}$ | $\mathbf{56 \pm 2}$ | $\mathbf{42 \pm 6}$ | $\mathbf{324 \pm 8}$ | $\mathbf{290 \pm 6}$ |
| FMCTS | $65 \pm 3$ | $76 \pm 3$ | $96 \pm 8$ | $499 \pm 10$ | $299 \pm 6$ |
| tmdp+DFS | $93 \pm 3$ | $204 \pm 3$ | $88 \pm 4$ | $521 \pm 10$ | $308 \pm 6$ |

In the Tab. 4, we show mean execution time and standard deviation computed for the same instance with different seeds and averaged over all task instances for the test tasks. All learning-based methods perform much faster than the Strong branching with execution time proportional to the number of nodes in B&B trees. To prove the statistical significance of our results, we perform paired difference test (Wilcoxon, 1945) between our method and baselines. Our null hypothesis is that TreeDQN performs similarly to our baselines, so the distribution of differences in execution times should be symmetric about zero. The results, shown at the bottom of Tab. 4, indicate strong evidence against the null hypothesis for almost all test tasks except Facility Location where TreeDQN performs close to IL and Multiple Knapsack where RL methods (TreeDQN, FMCTS, and tmdp+DFS) perform close to each other.

Table 4: Mean execution time with std averaged over five seeds for test tasks. W() denotes the Wilcoxon test between TreeDQN and methods in brackets.

| Model | Comb.Auct | Set Cover | Max.Ind.Set | Facility Loc. | Mult.Knap. |
|---|---|---|---|---|---|
| SCIP default | $2.19 \pm 32\%$ | $3.13 \pm 12\%$ | $4.77 \pm 20\%$ | $10.85 \pm 28\%$ | $0.63 \pm 54\%$ |
| Strong Branching | $4.89 \pm 11\%$ | $7.31 \pm 8\%$ | $127.09 \pm 26\%$ | $58.53 \pm 43\%$ | $9.96 \pm 104\%$ |
| IL | $0.84 \pm 7\%$ | $1.16 \pm 7\%$ | $2.18 \pm 12\%$ | $7.10 \pm 33\%$ | $7.90 \pm 113\%$ |
| TreeDQN | $\mathbf{0.86 \pm 8\%}$ | $\mathbf{1.16 \pm 8\%}$ | $2.72 \pm 14\%$ | $\mathbf{6.78 \pm 33\%}$ | $\mathbf{1.54 \pm 72\%}$ |
| FMCTS | $0.91 \pm 7\%$ | $1.42 \pm 8\%$ | $5.45 \pm 31\%$ | $12.26 \pm 39\%$ | $1.94 \pm 89\%$ |
| tmdp+DFS | $1.03 \pm 9\%$ | $2.29 \pm 15\%$ | $\mathbf{2.32 \pm 13\%}$ | $15.21 \pm 40\%$ | $2.44 \pm 86\%$ |
| W(IL) | $2.37 \cdot 10^{-9}$ | $1.08 \cdot 10^{-3}$ | $2.68 \cdot 10^{-4}$ | $2.66 \cdot 10^{-1}$ | $4.38 \cdot 10^{-9}$ |
| W(FMCTS) | $7.47 \cdot 10^{-5}$ | $1.61 \cdot 10^{-27}$ | $2.16 \cdot 10^{-26}$ | $1.44 \cdot 10^{-14}$ | $9.77 \cdot 10^{-1}$ |
| W(tmdp+DFS) | $1.34 \cdot 10^{-27}$ | $2.39 \cdot 10^{-34}$ | $5.00 \cdot 10^{-9}$ | $1.40 \cdot 10^{-13}$ | $5.28 \cdot 10^{-1}$ |

Another important metric is the gap between primal and dual bounds as a function of time shown in Fig. 3. In the Branch-and-Bound algorithm, the primal-dual gap monotonically decreases when solving an instance. The speed of the gap reduction is proportional to the number of nodes and mean execution time.

To further analyze the performance of our agent, we present distributions of tree sizes in the form of probability-probability plots (P-P plots) in Fig. 4. P-P plot allows us to compare different cumulative distribution functions (CDF). For a reference CDF $F$ and a target CDF $G$ P-P plot is constructed similar to the ROC curve: we choose a threshold $x$, move it along the domain of $F$ and draw points $(F(x), G(x))$. To show multiple distributions on the same plot, we use Strong Branching as reference CDF for all of them. If one curve is higher than another, then the corresponding CDF is larger, so

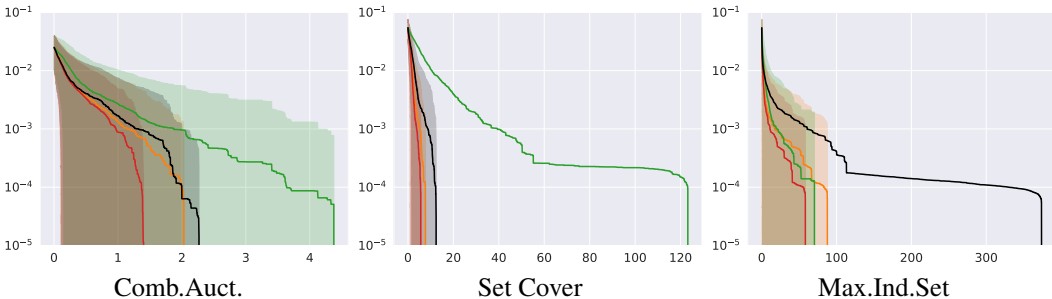

Figure 3: Primal-Dual gap as a function of time. Red - IL, orange - TreeDQN, black - FMCTS, green - tmdp+DFS.

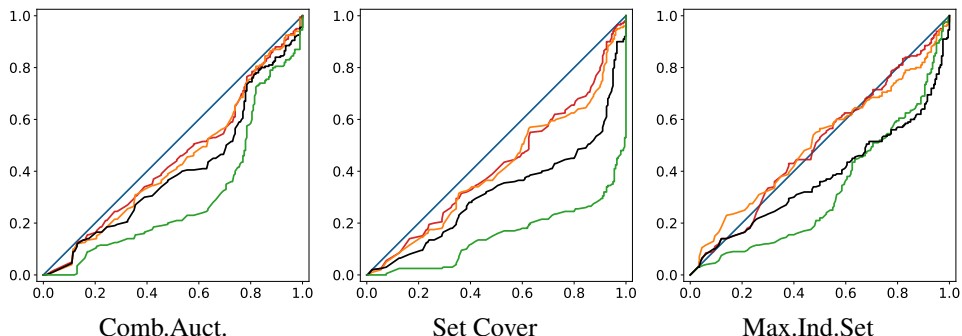

Figure 4: P-P plots of tree size distributions for test instances. Blue - Strong Branching, red - Imitation Learning, orange - TreeDQN, black - FMCTS, green - tmdp+DFS.

the associated agent can solve more tasks in $x$ or nodes or less. All our baselines (except Strong Branching) have close complexity per call. So if one curve is higher than another, the corresponding agent can solve more tasks at the same time. This is related to winning rates which shows the number of instances solved in a certain time limit (see Appendix B.2, Tab. 10, 11, 10). From the P-P plot for the Maximum Independent Set, we see that TreeDQN is good at solving simple tasks where it performs better than Imitation Learning. When the tasks become more complex, the performance of TreeDQN decreases. This behavior is the direct consequence of our learning objective. We optimize the geometric mean of expected tree size, so complex task instances may have less influence on the learning process. P-P plots and arithmetic means for all test tasks are shown in Appendix B.1, Fig. 7 and Tab. 8.

Besides testing the performance of our agent, we also study its abilities to generalize. Table 5 presents evaluation results for complex transfer tasks solved with five different seeds. Since solving complicated MILP problems is time-consuming, we limit the maximum number of nodes in a B&B tree to $200'000$. The number of transfer tasks terminated by this node limit is shown in Appendix B.1, Tab. 9. For terminated instances, we use node limit as tree size when computing the geometric mean. It is seen from Tab. 5 that in the Set Cover, Facility Location, and Multiple Knapsack tasks, our TreeDQN agent transfers well and performs better than the tmdp+DFS and FMCTS agents. In the Combinatorial Auction task, the FMCTS agent transfers slightly better than TreeDQN. In the Maximum Independent Set task, the TreeDQN agent falls behind the tmdp+DFS agent since it adapted better for simple task instances, as seen from the P-P plot 4.

## 6 ABLATION STUDY

In this section, we perform an ablation study and compare the performance of our agent with the TreeDQN agent trained with standard MSE loss function. Our modified learning objective prevents explosions of gradients and significantly stabilizes the training process as seen in Fig. 5 (Fig. 9 in Appendix C shows losses for all tasks). Training with smoother gradients should lead to a better

Table 5: Geometric mean with geometric std for transfer tasks.

| Model | Comb.Auct | Set Cover | Max.Ind.Set | Facility Loc. | Mult.Knap. |
|---|---|---|---|---|---|
| SCIP default | $672 \pm 5$ | $64 \pm 5$ | $914 \pm 4$ | $120 \pm 25$ | $5851 \pm 5$ |
| Strong Branching | $665 \pm 3$ | $122 \pm 4$ | $845 \pm 5$ | $722 \pm 14$ | $57639 \pm 4$ |
| IL | $867 \pm 3$ | $149 \pm 4$ | $2872 \pm 9$ | $608 \pm 8$ | $60530 \pm 4$ |
| TreeDQN | $1567 \pm 4$ | $\mathbf{174 \pm 4}$ | $4541 \pm 9$ | $\mathbf{759 \pm 11}$ | $\mathbf{35599 \pm 4}$ |
| FMCTS | $\mathbf{1375 \pm 3}$ | $252 \pm 4$ | $8647 \pm 9$ | $1135 \pm 11$ | $42461 \pm 5$ |
| tmdp+DFS | $2171 \pm 4$ | $858 \pm 6$ | $\mathbf{1713 \pm 5}$ | $847 \pm 10$ | $40316 \pm 5$ |

policy that can produce B&B trees with lower geometric mean. In Tab. 6, we compare the geometric mean of tree size for test instances for TreeDQN trained with MSLE and MSE loss functions. As seen from Tab. 6 in all tasks, the agent trained with a modified loss function achieves a lower geometric mean of the final tree size. Wilcoxon test (Wilcoxon, 1945) shown at the bottom of Tab. 6 indicates statistical significance that our modified loss function allows our agent to learn a better policy for the Combinatorial Auction, Set Cover, and Maximum Independent Set tasks.

Table 6: Geometric mean with geometric std for test tasks. MSE is a TreeDQN agent trained with MSE loss. W(MSE) denotes the Wilcoxon test between TreeDQN and MSE distributions.

| Model | Comb.Auct | Set Cover | Max.Ind.Set | Facility Loc. | Mult.Knap. |
|---|---|---|---|---|---|
| TreeDQN | $\mathbf{58 \pm 3}$ | $\mathbf{56 \pm 2}$ | $\mathbf{42 \pm 6}$ | $\mathbf{324 \pm 8}$ | $\mathbf{290 \pm 6}$ |
| MSE | $64 \pm 3$ | $58 \pm 2$ | $60 \pm 6$ | $352 \pm 8$ | $367 \pm 7$ |
| W(MSE) | $1.42 \cdot 10^{-9}$ | $2.29 \cdot 10^{-3}$ | $3.14 \cdot 10^{-4}$ | $1.15 \cdot 10^{-1}$ | $5.16 \cdot 10^{-2}$ |

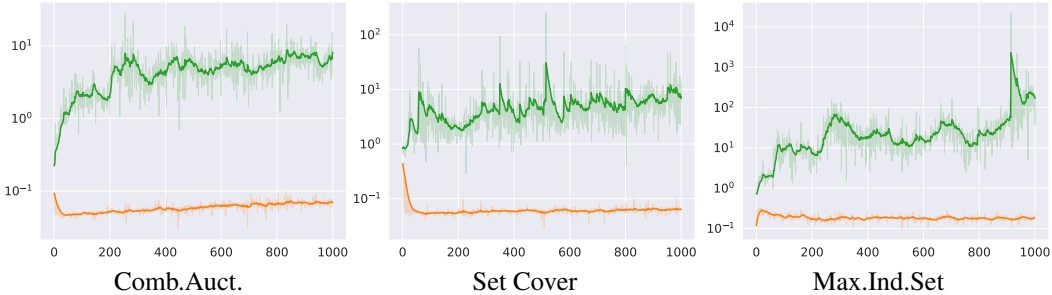

Figure 5: Loss as a function of number of episodes. Orange - TreeDQN with MSLE loss, green - with MSE loss.

# 7 CONCLUSION

We have presented a novel data-efficient deep reinforcement learning method to learn a branching rule for the Branch-and-Bound algorithm. The synergy of the exact solving algorithm and data-driven heuristic takes advantage of both worlds: guarantees to compute the optimal solution and the ability to adapt to specific tasks. Our method utilizes tree MDP and contraction property of the tree Bellman operator. It maps solving a MILP task to an episode for our RL agent and trains the agent to optimize the target metric — the resulting size of the B&B tree. We have proposed a modified learning objective that stabilizes the learning process in the presence of high variance returns. Our approach surpasses previous RL methods at all test tasks. The code is available at https://anonymous.4open.science/r/treedqn-C50A. In the future, we are interested in studying multitask branching agents and the limits of generalization to more complex task instances.

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

## A   COMPARISON OF REINFORCEMENT LEARNING METHODS USED FOR VARIABLE SELECTION

### A.1   ENVIRONMENT

To train TreeDQN, FMCTS, and tmdp+DFS we use a branching environment with the following properties:

**Observation.**   We use state representation in the form of a bipartite graph provided by Ecole (Prouvost et al., 2020). In this graph, edges correspond to connections between constraints and variables with weight equal to the coefficient of the variable in the constraint. Each variable and constraint node is represented by a vector of 19 and 5 features, respectively.

**Actions.**   The agent selects one of the fractional variables for splitting. Since the number of fractional variables decreases during an episode, we apply a mask to choose only among available variables.

**Rewards.**   At each step, the agent receives a negative reward $r = -1$. The total cumulative return equals the resulting tree size with a negative sign.

**Episode.**   In each episode, the agent solves a single MILP instance. We limit the solving time for one task instance during training to 10 minutes and terminate the episode if the time is over.

### A.2   LOSS FUNCTIONS

$$L_{tMDP+DFS} = - \log \pi_\theta(a_t|s_t) R(s_t) - \lambda H(\pi_\theta(\cdot|s_t)) \tag{7}$$

In eq. 7 $R(s_t)$ is cumulative discounted reward, $H(\pi_\theta(\cdot|s_t))$ - entropy of the policy $\pi_\theta(\cdot|s_t)$.

$$L_{FMCTS} = \left(\frac{Q(s_t, a_t) - R(s_t)}{size(root(s_t))}\right)^2 \tag{8}$$

In eq. 8 $size(root(s))$ is the size of tree which contains node $s_t$.

$$L_{TreeDQN} = (log(Q(s_t, a_t)) - log(r + Q_{target}(s_{t+1}^+, a_{t+1}^+) + Q_{target}(s_{t+1}^-, a_{t+1}^-)))^2 \tag{9}$$

In eq. 9 actions $a_{t+1}$ are computed as $a_{t+1}^{\pm} \leftarrow \arg\max_{a_{t+1}^{\pm}} Q(s_{t+1}^{\pm}, a_{t+1}^{\pm})$.

### A.3   TRAINING EFFICIENCY

Training parameters of the TreeDQN algorithm are shown in Tab. 7. In Fig. 6 we show the geometric mean of the tree size for validation instances during training. The checkpoint with the lowest geometric mean for each method was chosen for testing.

Table 7: Hyperparameters used in the training of the TreeDQN agent.

| parameter | value |
|---|---|
| $\gamma$ | 1 |
| buffer size | 100'000 |
| buffer min size | 1'000 |
| batch size | 32 |
| learning rate | $10^{-4}$ |
| $\varepsilon$-decay steps | 100'000 |
| number of training episodes | 1000 |
| optimizer | adam |

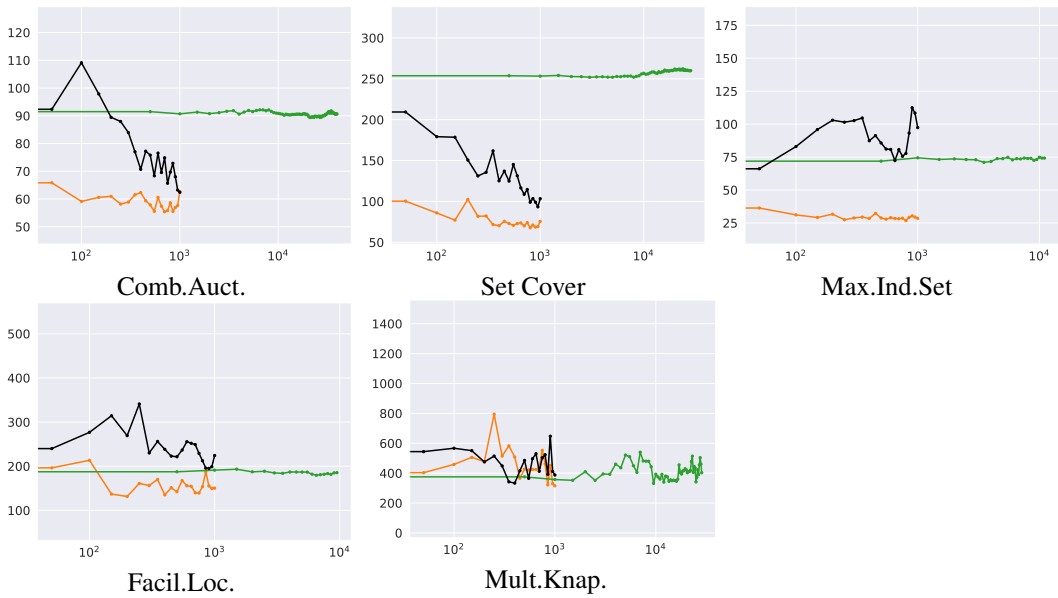

Figure 6: Tree size for validation instances as a function of number of training episodes. Orange - TreeDQN, black - FMCTS, green - tmdp+DFS.

## A.4 TREEDQN ALGORITHM

---

**Algorithm 1:** TreeDQN with experience replay

---

**Data:** Replay buffer capacity $N$, replay buffer minimum capacity $n$, discount factor $\gamma$, number of updates $t$, $\varepsilon$ decay function, batch size $b$, target update frequency $t_{up}$, random number generator R

**Initialize:** $Q_{\text{target}}, Q_{\text{net}}, \mathcal{D} \leftarrow \varnothing, \varepsilon \leftarrow 1$

**Result:** $Q_{\text{net}}$

$s \leftarrow$ env.reset();

**while** $i \leq t$ **do**

    **if** *R(0, 1) < $\varepsilon$* **then**

        a $\leftarrow$ random action;

    **else**

        a $\leftarrow \arg\max_a Q_{\text{net}}(s, a)$;

    **end**

    $s_{\text{next}}$, r, $s^+$, $s^-$ = env.step(a);

    $\mathcal{D} \leftarrow \mathcal{D} \cup (s, a, r, s^{\pm})$;

    $s \leftarrow s_{\text{next}}$;

    $\varepsilon \leftarrow$ decay($\varepsilon$);

    $i \leftarrow i + 1$;

    **if** $i > n$ **then**

        sample batch $(s, a, r, s^{\pm}) \sim \mathcal{D}$;

        $a^{\pm} \leftarrow \arg\max_{a^{\pm}} Q_{\text{net}}(s^{\pm}, a^{\pm})$;

        target $= \log\left[|r + \gamma(Q_{\text{target}}(s^+, a^+) + Q_{\text{target}}(s^-, a^-))|\right]$;

        loss $= \frac{1}{b} \sum (\log(|Q_{\text{net}}(s, a)|) - \text{target})^2$;

        $Q_{\text{net}} \leftarrow$ optimize($Q_{\text{net}}$, loss)

    **end**

    **if** $i \mod t_{up}$ *= 0* **then**

        $Q_{\text{target}} \leftarrow Q_{\text{net}}$

    **end**

**end**

---

# B EVALUATION

## B.1 TREE-SIZE METRICS

In Fig. 7 we show probability-probability plots of distributions of tree sizes for the test instances. In Tab. 8 we show the mean of distributions of tree sizes for test tasks. In Tab. 9 we show a number of transfer tasks that reached the node limit of 200'000 nodes and were stopped.

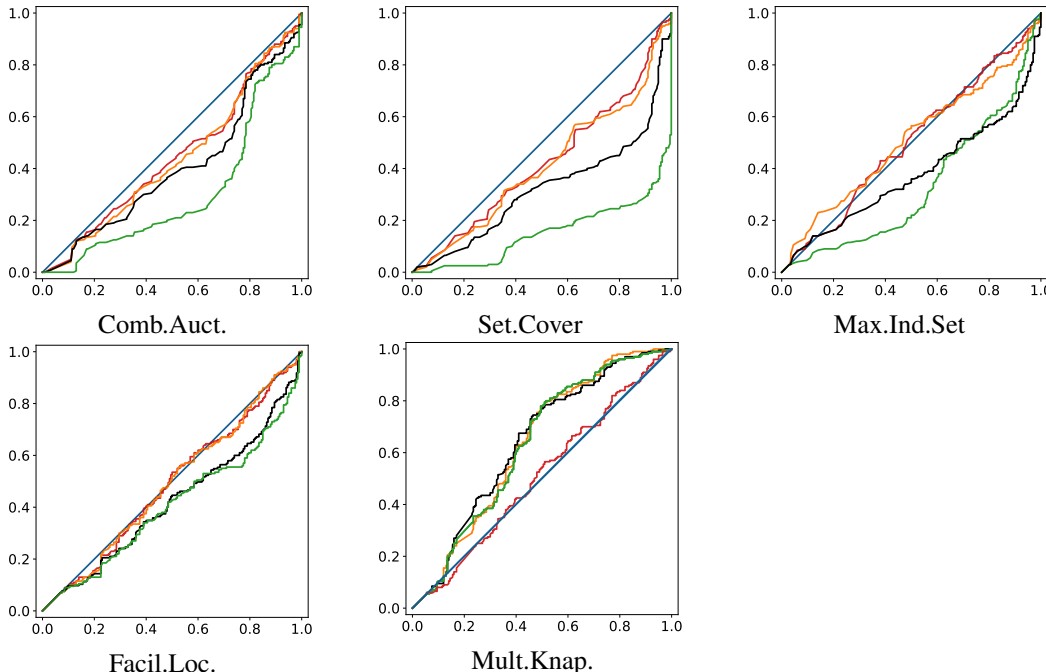

Figure 7: P-P plots of tree size distributions for test instances. Blue - Strong Branching, red - Imitation Learning, orange - TreeDQN, black - FMCTS, green - tmdp+DFS.

Table 8: Mean with std averaged over 5 seeds for test tasks.

| Model | Comb.Auct | Set Cover | Max.Ind.Set | Facility Loc. | Mult.Knap. |
|---|---|---|---|---|---|
| SCIP default | $18 \pm 41\%$ | $25 \pm 23\%$ | $89 \pm 48\%$ | $290 \pm 61\%$ | $496 \pm 76\%$ |
| Strong Branching | $76 \pm 14\%$ | $59 \pm 8\%$ | $126 \pm 36\%$ | $1035 \pm 53\%$ | $5139 \pm 116\%$ |
| IL | $91 \pm 14\%$ | $71 \pm 9\%$ | $194 \pm 32\%$ | $1113 \pm 46\%$ | $4243 \pm 120\%$ |
| TreeDQN | $\mathbf{95 \pm 14\%}$ | $\mathbf{78 \pm 10\%}$ | $308 \pm 36\%$ | $\mathbf{1118 \pm 44\%}$ | $\mathbf{852 \pm 86\%}$ |
| FMCTS | $113 \pm 15\%$ | $114 \pm 14\%$ | $1036 \pm 61\%$ | $1991 \pm 50\%$ | $1231 \pm 107\%$ |
| tmdp+DFS | $160 \pm 18\%$ | $405 \pm 24\%$ | $\mathbf{255 \pm 42\%}$ | $2409 \pm 50\%$ | $1457 \pm 103\%$ |

Table 9: Number of transfer tasks finished by node limit. The total number of instances with different seeds is 200.

| Model | Comb.Auct | Set Cover | Max.Ind.Set | Facility Loc. | Mult.Knap. |
|---|---|---|---|---|---|
| SCIP default | 0 | 0 | 0 | 0 | 2 |
| Strong Branching | 0 | 0 | 0 | 10 | 42 |
| IL | 0 | 0 | 5 | 0 | 44 |
| TreeDQN | 0 | 0 | 4 | 6 | 23 |
| FMCTS | 0 | 0 | 25 | 1 | 37 |
| tmdp+DFS | 0 | 0 | 1 | 2 | 36 |

## B.2 RUN-TIME METRICS

In Fig. 8 we show primal-dual gap as a function of time.

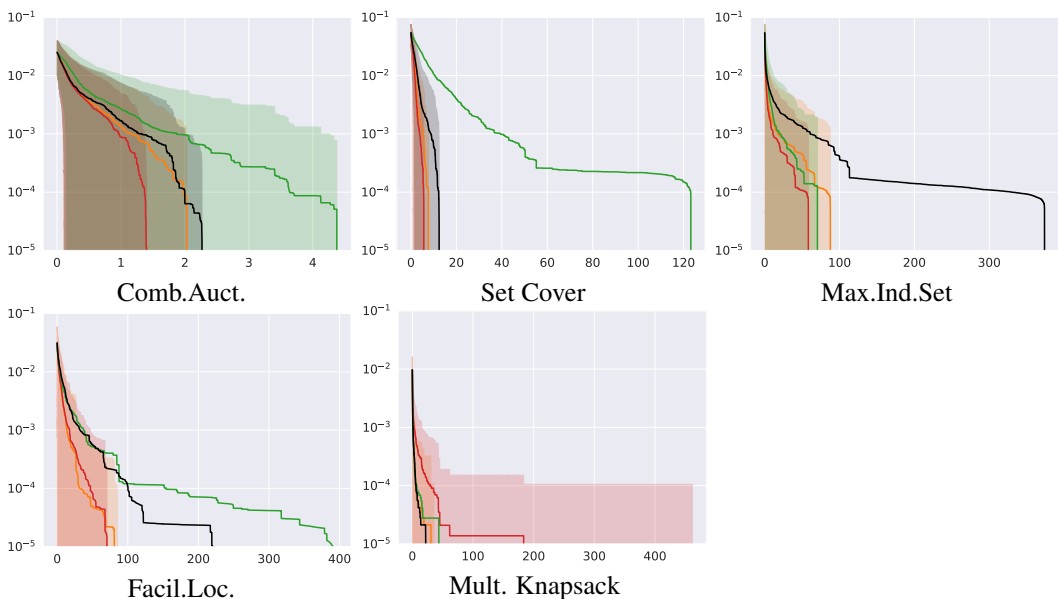

Figure 8: Primal-Dual gap as a function of time. Red - Imitation Learning, orange - TreeDQN, black - FMCTS, green - tmdp+DFS.

In addition to the primal-dual gap plots in Tab. 10, 11, 12 we provide winning rates - the number of instances which were solved down to optimal solution at the same time as IL agent solves 25%, 50%, 75% and 100% of tasks.

Table 10: Winning rates: Combinatorial auction and Set Cover tasks.

| Model | Comb.Auct | | | | Set Cover | | | |
|---|---|---|---|---|---|---|---|---|
| | 25% | 50% | 75% | 100% | 25% | 50% | 75% | 100% |
| SCIP default | 0.00% | 0.00% | 0.00% | 22.00% | 0.00% | 0.00% | 0.00% | 7.50% |
| Strong Branching | 0.00% | 0.00% | 0.00% | 20.00% | 0.00% | 1.50% | 2.00% | 12.50% |
| IL | 25.00% | 50.00% | 75.00% | 100.00% | 25.00% | 50.00% | 75.00% | 100.00% |
| TreeDQN | 21.50% | 47.50% | 73.50% | 96.50% | 21.00% | 53.00% | 72.50% | 97.50% |
| FMCTS | 23.00% | 43.50% | 70.00% | 94.50% | 17.50% | 36.50% | 51.50% | 93.00% |
| tmdp+DFS | 16.50% | 28.50% | 61.50% | 92.50% | 6.50% | 23.50% | 35.00% | 69.00% |

Table 11: Winning rates: Maximum Independent Set and Facility Location tasks.

| Model | Max.Ind.Set | | | | Facil. Loc. | | | |
|---|---|---|---|---|---|---|---|---|
| | 25% | 50% | 75% | 100% | 25% | 50% | 75% | 100% |
| SCIP default | 0.00% | 0.00% | 0.00% | 99.00% | 7.00% | 16.00% | 42.50% | 100.00% |
| Strong Branching | 0.00% | 0.00% | 0.00% | 11.86% | 2.15% | 4.84% | 18.82% | 69.89% |
| IL | 25.00% | 50.00% | 75.00% | 100.00% | 25.00% | 50.00% | 75.00% | 100.00% |
| TreeDQN | 29.38% | 53.09% | 71.65% | 97.42% | 24.73% | 48.92% | 79.57% | 99.46% |
| FMCTS | 21.13% | 32.99% | 54.12% | 95.36% | 20.43% | 38.71% | 60.22% | 97.85% |
| tmdp+DFS | 10.82% | 31.44% | 57.22% | 100.00% | 20.97% | 43.55% | 61.29% | 97.85% |

Table 12: Winning rates: Multiple Knapsack task.

| Model | 25% | 50% | 75% | 100% |
|---|---|---|---|---|
| SCIP default | 43.00% | 94.00% | 100.00% | 100.00% |
| Strong Branching | 16.93% | 48.15% | 74.60% | 99.47% |
| IL | 25.40% | 50.26% | 75.13% | 100.00% |
| TreeDQN | 42.33% | 76.72% | 93.12% | 100.00% |
| FMCTS | 44.97% | 76.19% | 91.53% | 100.00% |
| tmdp+DFS | 34.39% | 76.19% | 93.12% | 100.00% |

## C  ABLATION STUDY

We show a loss as a function of number of training episodes in Fig. 9.

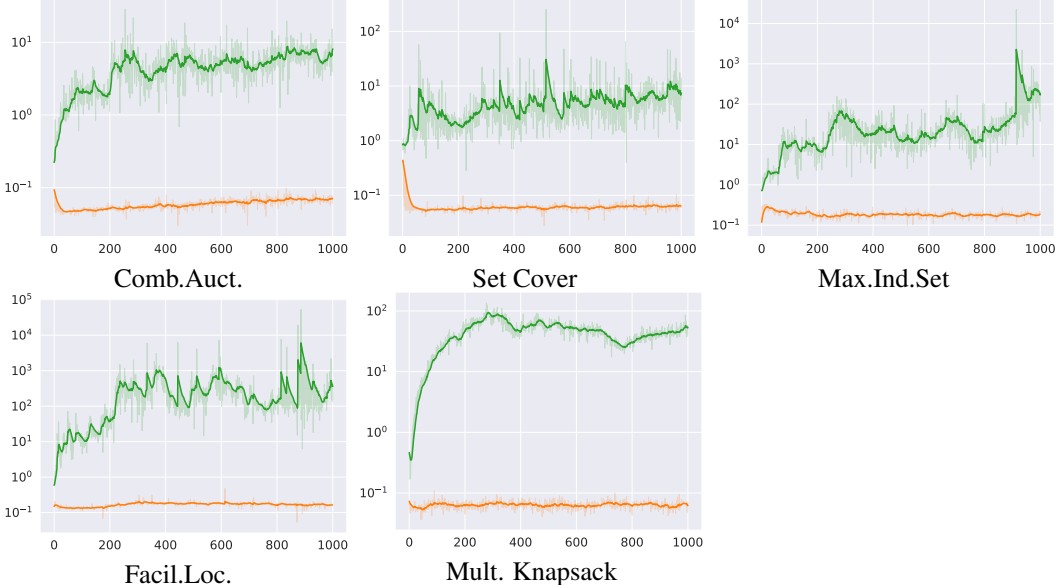

Figure 9: Loss as a function of number of episodes. Orange - TreeDQN with MSLE loss, green - TreeDQN with MSE loss.

## D  BALANCED ITEM PLACEMENT

We evaluate our method on a challenging Balanced Item Placement dataset (ML4CO competition, Gasse et al. (2022)). In the dataset, problem instances are modeled as multi-dimensional multi-knapsack MILP tasks. Each task represents the spreading of items across containers, such as spreading files across disks or distributing processes across different machines with even utilization. The number of movable items is constrained to model the real-life situation of a live system. The dataset contains 9900 train instances, 100 validation, and 100 test instances.

### D.1  ENVIRONMENT

**Observations and actions.**  The agent observes a bipartite graph and returns an index of a variable for splitting.

**Rewards.**  We train the agent to maximize the dual integral. The dual integral measures the area under the curve of the solver's global lower bound (dual bound), which corresponds to a solution of LP relaxation of the MILP. When the agent chooses branching variables, the domain of integer

variables gets tightened, and the dual bound increases over time. The dual integral is defined as follows:

$$I_d = \int_{t=0}^{T} z_t^* dt,$$

where $T$ is the time limit, and $z_t^*$ is the best dual bound found at time $t$. At each time step, the agent receives the reward equal to the dual integral since the previous state, so the cumulative return is equal to the dual integral $I_d$.

**Episode.** In each episode, the agent solves a single MILP instance. The episode duration is limited to 15 minutes during both training and evaluation.

## D.2 TRAINING

We train our agent with the same hyper-parameters and the same architecture as in previous tasks. Since each episode in this environment takes 15 minutes to complete we decrease the number of training episodes to 500. Fig. 10 shows the loss as a function of number of updates. The loss decreases during training as our agent learns to predict returns better.

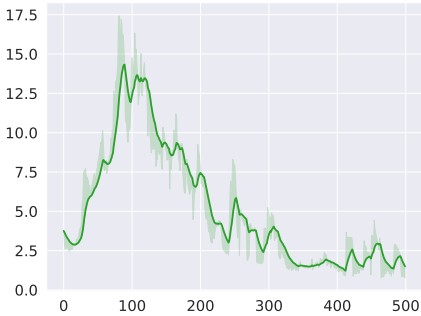

Figure 10: Loss as a function of number of episodes.

This environment highlights the sample efficiency of our method because training on-policy methods in this environment would be problematic.

## D.3 EVALUATION RESULTS

We compare the performance of the TreeDQN agent with the SCIP solver, Strong Branching heuristic, and Imitation Learning agent on 100 test instances. Since the tasks are complex, all tasks for each branching method were finished by reaching the 15-minute time limit. We present evaluation results in Tab. 13 and Tab. 14. It is seen from Tab. 13 that the TreeDQN agent achieves the highest cumulative reward by a significant margin. Comparing the TreeDQN and IL agents, which use the same GCNN architecture, we see that for the same amount of time, TreeDQN solves significantly fewer LP tasks. This is because it creates more complex LPs which increase the dual bound faster.

Table 13: Evaluation on balanced item placement task.

| Model | Reward | # Nodes $\times 10^3$ | # LPs $\times 10^3$ | Primal bound | Dual bound |
|---|---|---|---|---|---|
| SCIP default | 3885.24 | 258.36 | 5037.10 | 18.46 | 4.97 |
| Strong Branching | 3419.00 | 0.552 | 13.95 | 628.02 | 4.01 |
| IL | 4964.77 | 141.36 | 1911.16 | 537.85 | 5.92 |
| TreeDQN | 5958.06 | 83.76 | 846.40 | 87.33 | 7.05 |

We present quantiles of the dual bound distributions in Tab. 14. We see that the dual bound distribution of the TreeDQN agent has larger 50%, 75%, and 100% quantiles than the distributions of other methods.

Table 14: Dual bound distribution.

| Model | q = 25% | q = 50% | q = 75% | q = 100% |
|---|---|---|---|---|
| SCIP default | 1.18 | 3.76 | 7.24 | 20.88 |
| Strong Branching | 1.72 | 3.54 | 5.88 | 12.95 |
| IL | 3.65 | 5.18 | 7.93 | 17.53 |
| TreeDQN | 3.37 | 5.59 | 9.37 | 23.27 |

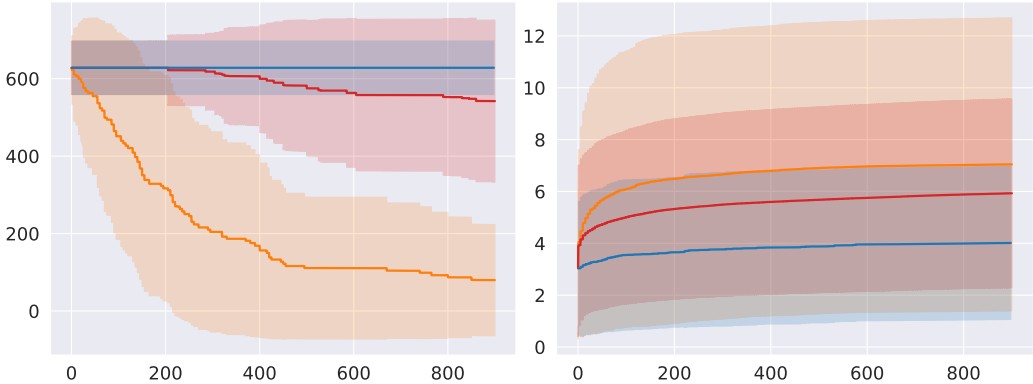

Figure 11: Primal bound (on the left) and dual bound (on the right) as a function of time. Red - Imitation Learning, orange - TreeDQN, blue - Strong Branching.

Fig. 11 shows the primal and dual bounds as a function of time. The TreeDQN agent decreases the primal bound and increases the dual bound much faster than the IL and Strong Branching agents. In conclusion, the evaluation results demonstrate that the TreeDQN agent learns an effective branching policy, is sample-efficient, and can be trained with only 500 training episodes.

