# OpenReview forum: "TreeDQN: Learning to minimize Branch-and-Bound tree"
_ICLR.cc/2024/Conference — Submitted to ICLR 2024_

### Official Review · Reviewer_vJVo · 2023-10-24

**Soundness:** 3 good
**Presentation:** 3 good
**Contribution:** 3 good
**Rating:** 6
**Confidence:** 3

**Summary:**

This paper extended the on-policy learning to branch method introduced by Scavuzzo et al. in 2022 to an off-policy setting by offering a proof of contraction in mean, a modified mean squared logarithmic error, and an adapted Double Dueling DQN scheme.

**Strengths:**

1. The evaluation experiments demonstrate a noteworthy improvement compared to previous work and other state-of-the-art approaches.

2. The modified mean squared logarithmic error proves to be well-suited for long-tailed distributions of BB tree sizes and exhibits superior performance compared to the mean squared error in the ablation study.

**Weaknesses:**

My main concerns about this paper are generalization ability, scalability, and some basic assumptions. Please find details in the questions.

**Questions:**

1. Regarding the Assumption in Theorem 4.1: The paper assumes that the probability of having left and right children does not depend on the state because the pruning decision depends on the global upper bound instead of the parent node. However, the global upper bound can change dynamically during the search, which might influence the probability. Does this paper use optimal solutions as upper bounds? Could the authors provide further clarification on this assumption?

2. Exploring Limited Generalization Ability: In comparing the results presented in Table 5 and Table 3, it is observed that TreeDQN appears to exhibit less stability in the context of transfer tasks. Could you please offer insights or explanations regarding this phenomenon?

3. A Traditional vs. RL-based Variable Selection Perspective: Traditional variable selection methods rely on human-designed criteria, such as pseudocosts. One advantage of these traditional approach is its applicability to various problem types. On the other hand, current RL-based methods require training an optimal policy for each specific problem. Given the noted limitations in generalization ability, RL methods seem to necessitate training on problem instances of a similar size as the target problems. Could you provide any comments or insights on the potential implications of this limitation? (This question is optional, and your input is welcomed purely out of curiosity.)

---

> ### Author Response · Authors · 2023-11-21
> **Thank you for your review**
>
> > However, the global upper bound can change dynamically during the search, which might influence the probability. Does this paper use optimal solutions as upper bounds? Could the authors provide further clarification on this assumption?
>
> In our experiments, we use DFS node selection during training and switch to the default node selection strategy during testing. We also experimented with training using the default node selection strategy. We observed that the agent trained with the default node selection strategy performs slightly less, compared to our final setup, due to higher stochasticity and violation of the Markov property of the training environment.
>
> We present Theorem 4.1 to support the statement that a reinforcement learning agent should be able to converge to an optimal policy in a fully observed tree MDP when the probabilities  $p^+$ and $p^-$ do not depend on the state. Indeed, the branching process in the actual B&B solver may violate some initial assumptions like the Markov property or the independence of $p^+$, $p^-$ probabilities from the state. However, if the state does not contain the exact value of the global upper bound, we can not accurately predict the number of child nodes. Thus, we can consider probabilities $p^+$ and $p^-$ independent from the state to some extent. So, if the violation of initial assumptions is small enough, the agent should be able to converge to a well-performing policy, as we demonstrate in our experiments.
>
> > Exploring Limited Generalization Ability: In comparing the results presented in Table 5 and Table 3, it is observed that TreeDQN appears to exhibit less stability in the context of transfer tasks. Could you please offer insights or explanations regarding this phenomenon?
>
> Thank you for raising an interesting question! We train the TreeDQN agent to optimize the geometric mean of the expected return. So rare, large trees may have a less significant impact on the final policy. It can be seen from the Probability-probability plot for the Maximum Independent Set task (Fig. 4) that the TreeDQN starts falling behind the IL agent when the tasks become harder (in the upper right corner). Thus, in complex transfer tasks, the TreeDQN agent may underperform.
>
> > A Traditional vs. RL-based Variable Selection Perspective
>
> RL methods can find a new branching strategy that could perform better than the traditional methods for a specific distribution of tasks. We believe that RL methods can improve the performance of Branch-and-Bound solvers in areas that require frequent solving of similar MILPs. For example, when a logistics company ships goods to customers or when we need to allocate limited resources like human workers or computing resources of a data center. At the present time, human-designed heuristics are better for tasks when you need to solve problems with significantly varying complexity and type or have only several task instances of the same kind.  However, the development of multitask and meta-learning approaches may extend the applicability of reinforcement learning methods.

---

### Official Review · Reviewer_X3jH · 2023-10-29

**Soundness:** 2 fair
**Presentation:** 2 fair
**Contribution:** 3 good
**Rating:** 3
**Confidence:** 4

**Summary:**

This work studies the variable selection problem in the branch-and-bound algorithm from the point of view of Tree-MDPs, which, instead of the “linear” time-axis present in ordinary markov decision processes, models the decision history as a binary tree.
They show that under mild assumptions tree-MDPs allow for a contractive Bellman operator, justifying a Tree-MDP version of deep q-learning dubbed TreeDQN. Finally, the authors demonstrate their performance against the “strong branching” baseline and other learnt variable selectors on a large set of synthetic instances.

**Strengths:**

Inherently, the idea of modelling variable selection as a Tree-MDP is a great idea as it allows the incorporation of the branch-and-bound structure into the decision process. The modification of the loss function to stably regress towards the geometric mean is also clever and might prove useful even outside the learnt variable selection domain. In general, the presentation of the work is clean and easy to read.

**Weaknesses:**

1. Perhaps the biggest limitation is the assumption that the upper bound has to be derivable from the current node or known ahead of time. The authors assert that this (as well as more intricate node selection policies) lead to at most a moderate distribution shift, but never demonstrate this effect.
2. Another concern is regarding the difficulty distribution of instances. Random instance generation has been known to generate significant amounts of trivial instances compared to real-world equivalents. However, this is a limitation of most prior work on learnt variable selection rules as well.
3. TreeDQN is also more expensive in terms of wall-clock-time than prior work (especially the IL agent), which can be seen in Figure 4. The paper does not make it clear whether this is due to TreeDQN using a different architecture, or TreeDQN simply creating more expensive nodes during branching.
4. An important missing baseline in their comparisons is out-of-the-box SCIP, acting as an automatic state-of-the-art hand-crafted tradeoff between SB and cheaper heuristics.


The paper needs an extensive re-write in terms of argumentation and clarity.


Some more points:
- Abstract: BnB solver[s] split a task…
- Abstract: …the Bellman operator adapted for the tree MDP is contracting in mean… - initially I did not understand what you mean with that (only at some later point into the paper)
- Intro: with [the] Branch-and-Bound algorithm (B&B). |[The] B&B algorithm employs…
- “The variable selection process is the most computationally expensive and crucial for the performance of the whole algorithm” – is there a reference to prove this? If not, omit this sentence
- Intro: “problematic”  challenging
- Intro: “single next state [the] agent”
- Intro: the contribution list at the end of the section looks like a draft and comes out of nothing
- Sec. 2: where objective… sentence broken
- Sec. 2: B&B [-algorithm-] builds
- Sec. 2: explain “relaxed”
- Sec. 2: Fig. 1 does not bring much to the table. I suggest to explain B&B with Fig. 1 right from the beginning (add primal/dual, relaxation, variables). This does not cost more space but helps to understand B&B
- Sec. 2.: [A] straight forward strategy
- Sec. 2.: [The] tree MDP was proposed by … In the tree MDP [the] value…
- Sec. 2.: The variable selection process … this paragraph is hard to understand
- Sec. 3.: “Our work improves…” please add some (technical) argument why this is the case
- Sec. 4.0: this part takes much space and can be omitted imho. Instead focus on explaining the bullet-point list at the end of 4.0 in more detail. Why must a successful RL method should have off-policy as a property? Policy gradient methods are great, and they are on-policy… Here are a lot of arguments that need more justification.
- Sec. 4.1 [E]quation3, [E]quation 4
- Sec. 4.1 is not satisfying to me. The section and with an inequality and tells me that the proof follows from the fact that the tree is finite. Please work out this prove in more detail.
- Sec. 4.2 the loss function [from E]quation 5
- Sec. 4.3 we use loss function equation 5 – please re-write
- Fig. 3 put the description into the plots

**Questions:**

- Is the method run on CPU or GPU?
- What is the performance of SCIP with default parameters on these instances (I.e. reliability pseudocost branching)?
- What is the model architecture (or more importantly: is it the same for all methods)?

---

> ### Author Response · Authors · 2023-11-21
> **Thank you for your review**
>
> > Perhaps the biggest limitation is the assumption that the upper bound has to be derivable from the current node or known ahead of time. The authors assert that this (as well as more intricate node selection policies) lead to at most a moderate distribution shift, but never demonstrate this effect.
>
> In our experiments we use DFS during training and switch to the default node selection strategy during evaluation. If we assume that the Strong Branching heuristic is close to optimal in our benchmark tasks (Combinatorial Auction, Set Cover, Maximum Independent Set, Facility Location), we can estimate the maximum value of possible performance degradation as the difference between the performance of TreeDQN and Strong Branching:
>
> Combinatorial Auction - 22%, Set Cover - 30%, Maximum Independent Set - 5%, Facility Location - 10%.
>
> > Another concern is regarding the difficulty distribution of instances. Random instance generation has been known to generate significant amounts of trivial instances compared to real-world equivalents. However, this is a limitation of most prior work on learnt variable selection rules as well.
>
> In our work, we focus on the same tasks as previous works (Gasse, NeurIPS 2020, Scavuzzo, NeurIPS 2022) to fairly benchmark our method with methods from the literature. In the updated version of our paper, we added a more challenging Balanced Item Placement task (see general response and updated paper, Appendix D).
>
> > TreeDQN is also more expensive in terms of wall-clock-time than prior work (especially the IL agent), which can be seen in Figure 4. The paper does not make it clear whether this is due to TreeDQN using a different architecture, or TreeDQN simply creating more expensive nodes during branching.
>
> Thank you for spotting this. We use the same Graph Convolutional Neural Network encoder architecture for all models in our benchmarks (IL, tmdp+DFS, FMCTS, TreeDQN). The TreeDQN is creating more expensive nodes during branching. It is clearly seen from the evaluation results on the Balanced Item Placement task (see Appendix D). In this task, all problem instances were finished by reaching a timeout. The TreeDQN agent solves much fewer LPs but achieves a much higher reward and much lower primal bound than the IL agent.
>
> > An important missing baseline in their comparisons is out-of-the-box SCIP, acting as an automatic state-of-the-art hand-crafted tradeoff between SB and cheaper heuristics.
>
> We added SCIP to our evaluation results. Please see the general response and updated paper.
>
> > The paper needs an extensive re-write in terms of argumentation and clarity.
>
> Thank you for providing detailed feedback! We corrected the issues. Please see the updated version of our paper.
>
> > Is the method run on CPU or GPU?
>
> GPU
>
> > What is the performance of SCIP with default parameters on these instances (I.e. reliability pseudocost branching)?
>
> Please see updated version of the paper
>
> > What is the model architecture (or more importantly: is it the same for all methods)?
>
> We use the same Graph Convolutional Neural Network encoder architecture for all models in our benchmarks (IL, tmdp+DFS, FMCTS, TreeDQN).

---

### Official Review · Reviewer_ZMqh · 2023-11-01

**Soundness:** 2 fair
**Presentation:** 2 fair
**Contribution:** 3 good
**Rating:** 3
**Confidence:** 3

**Summary:**

The authors use the TreeMDP framework introduced by Scavuzzo et al. to study RL methods for improved variable selection/branching in branch-and-bound for integer programming with the ultimate goal being smaller search trees. They propose a more stable and sample efficient RL training procedure by choosing a loss function to minimize the geometric mean of tree size during training, and use a deep Q network for training rather than the REINFORCE method used by Scavuzzo et al.

**Strengths:**

Branching is a critical aspect of integer programming solvers, and the authors provide an interesting new contribution towards RL based methods for the design of branching rules. The new methods are shown to produce smaller branch-and-bound trees than previous RL based variable selection methods, making this work a promising advance in the “learning to branch” line of work.

**Weaknesses:**

Section 2.2 “Tree MDP” needs way more explanation. It more or less assumes familiarity with the Tree MDP work of Scavuzzo et al., and a more self-contained exposition would be very helpful.

The theoretical contribution is very hazy to me. Contraction in mean is not really well-motivated. Does the cited theorem (Jaakkola ‘93) apply to the setting of tree operators here? That seems like a nontrivial assumption that is missing justification. Rather than just including a theorem about contraction in mean, the authors should have a main theorem that states the actual convergence guarantee that follows.

My understanding is that this paper is methodologically very similar to Scavuzzo et al., and only differs in the mechanics of how the RL algorithm is trained. This is discussed in Sections 4.2 and 4.3. In Section 4.2, the main difference is that the authors use a loss function that appears to be selectively picked based on the objective of minimizing the geometric mean of the tree sizes during training/testing. This to me feels like a specific and brittle design choice.

The new method is shown to yield smaller branch-and-bound trees than previous RL based variable selection policies, but no comparison is made to the default settings of any state-of-the-art solver (e.g., Gurobi, CPLEX, SCIP). This is an important comparison that should be included.

Overall the presentation did not convince me that this is a sufficiently novel contribution for ICLR. It seems like the authors just slightly tweaked some aspects of the methodology of Scavuzzo et al. It’s great that these modifications work and yield promising experimental results, but I just did not find the current writeup to be a sufficiently original contribution. The writeup itself also needs quite a bit of work to make it a cohesive, readable, and self-contained (the theory is presented in a very ad-hoc manner without formal definitions) contribution.

**Questions:**

“In the B&B search trees, the local decisions impact previously opened leaves via fathoming due to global upper-bound pruning. Thus the credit assignment in the B&B is biased upward, which renders the learned policies potentially sub-optimal.” I understand the first sentence, but what does the second sentence mean? What is “credit assignment”, and why is it/what does it mean for it to be biased upward?

See also questions in the “weaknesses” section.

---

> ### Author Response · Authors · 2023-11-21
> **Thank you for your review**
>
> > Section 2.2 “Tree MDP” needs way more explanation.
>
> We updated the background section and extended the description of Tree MDP. Please see the updated version.
>
> >  Rather than just including a theorem about contraction in mean, the authors should have a main theorem that states the actual convergence guarantee that follows.
>
> The convergence of Q-learning methods can be rigorously proved only for tabular MDPs, which is out of the scope of the present paper. To provide an intuition as to why our method should work, we prove the contraction in mean property of the tree Bellman operator.
>
> > My understanding is that this paper is methodologically very similar to Scavuzzo et al., and only differs in the mechanics of how the RL algorithm is trained.
>
> Our method has two significant differences from the work of Scavuzzo et.al.:
> 1. Use of MSLE loss function. Distribution of tree sizes in the B&B method will always have a long tail (if we can not prune a tree, it will be a complete binary tree with the size growing exponentially with its depth). The MSLE loss function should generally work better than the MSE when predicting the size of the tree because it is more stable to outliers.
> 2. Our method is much more sample efficient than the method of Scavuzzo et.al.(tmdp+DFS). To make one gradient update tmdp+DFS needs to solve a batch of MILP tasks until completion. In our approach, we make as many gradient updates as the size of the tree. To further prove the quality of our method, we tested it on a challenging Balanced Item Problem from the ML4CO competition (NeurIPS 2022, see Appendix D). We show that our TreeDQN agent can get a mean reward higher than the Imitation learning baseline. Since solving a single MILP task in this problem takes 15 minutes, training a tmdp+DFS in a reasonable amount of time would be impossible.
>
> > No comparison is made to the default settings of any state-of-the-art solver (e.g., Gurobi, CPLEX, SCIP)
>
> We added evaluation for the SCIP solver with the default set of parameters. Initially, we did not include it because the internal branching rules can make various modifications to the state of the solver, so it can not be considered a direct competitor to other methods. For more information see:
> 1. Discussion by Maxim Gasse (https://github.com/ds4dm/ecole/discussions/286#discussioncomment-2317487)
> 2. Gerald Gamrath and Christoph Schubert. Measuring the impact of branching rules for mixed-integer programming. In Operations Research Proceedings 2017, pages 165–170. Springer, 2018.
>
> > Overall the presentation did not convince me that this is a sufficiently novel contribution for ICLR.
>
> We updated our paper to address the issues you mentioned. We would like to emphasize that our method is significantly more sample efficient than the method of Scavuzzo et.al. When solving a single MILP task, we can perform N gradient updates where N is the size of the resulting B&B tree, while tmdp+DFS needs to solve a batch of MILPs to make a single gradient update. This results in a much higher sample efficiency of our method. To further demonstrate the sample efficiency of our method,  we trained and evaluated TreeDQN on the challenging Balanced Item Placement task from the ML4CO competition. The results in Appendix D demonstrate that our method outperforms Imitation Learning and Strong Branching in terms of total reward by a high margin. This result will not be possible with less sample-efficient methods since solving a single MILP instance of this problem requires 15 minutes.
>
> >  I understand the first sentence, but what does the second sentence mean? What is “credit assignment”, and why is it/what does it mean for it to be biased upward?
>
> Credit assignment is a problem in reinforcement learning when the immediate actions of an agent lead to rewards in the distant future, and the agent needs to figure out the connection between its actions and the delayed reward signal. We agree that this sentence is not clear and changed it to:
>
> In the B&B search trees, the local decisions impact previously opened leaves via fathoming due to global upper bound pruning, which violates the Markov property.

---

### Official Review · Reviewer_Uoqv · 2023-11-01

**Soundness:** 2 fair
**Presentation:** 1 poor
**Contribution:** 2 fair
**Rating:** 5
**Confidence:** 3

**Summary:**

This paper introduces TreeDQN, a reinforcement learning algorithm based on DQN for solving Tree MDPs. TreeDQN is trained on the mean squared logarithmic error loss. Specifically, the algorithm is used to learn branching heuristics for branch and bound in the context of mixed integer linear programming problems.

Empirical results on a set of benchmark problems show some of the advantages of TreeDQN for the purpose of learning a branching heuristic. The results on unseen tasks are somewhat mixed, with some advantage to the branching heuristic learned with TreeDQN.

**Strengths:**

The paper presents an algorithm for solving Tree MDPs with the specific application to learning branching heuristics for branch and bound algorithms in the context of solving mixed integer linear programming problems. TreeDQN presents better results on some of the benchmark problems used in the paper.

**Weaknesses:**

The presentation is *possibly* the paper's weakest point. The lack of clarity makes me wonder about the value of the value of the contributions of the paper. The main contribution of the paper, TreeDQN, is explained in a single paragraph in the main text. Since the text only states that the algorithms is an adaptation of Double Dueling DQN, I assume TreeDQN is a straightforward adaption of DQN to Tree MDPs.

The paper builds on a couple of previous papers, which I had to skim over in order to understand the present paper. I am not entirely familiar with the line of work of using RL to learn how to branch and I can tell that the paper wasn't written for me. These are the two papers that helped me understand this submission:

Exact Combinatorial Optimization with Graph Convolutional Neural Networks
and
Learning to Branch with Tree MDPs

The example on Mixed Integer Linear Programming isn't very helpful. The tree shown in Figure 1 is uninformative; it simply shows nodes in a tree where the color scheme differs the root of the inner nodes and from some of the leaf nodes. It would have been more helpful to not show a tree and give the reader a full example on how the branch and bound search works. I asked ChatGPT for an example and it gave me an example (without any drawings, of course) that was more helpful than the tree example shown in the paper.

Overall the background section could be re-written to use less space and pack more information to help the reader understand the work.

I cannot understand the last paragraph of Section 2.2 without reading the paper by Scavuzzo et al. (2022). Here are the question I asked myself while reading that paragraph.

1. Why do we need to use DFS as node selection or set the global upper bound in the root to the optimal solution cost to guarantee the Markov property?
2. The gap between training and testing is due to assuming that one has the optimal solution in training? Why not use DFS and not assume that you have the optimal solution in training?
3. How can more efficient heuristics for node selection also induce a gap between training and testing? And why is this important?

Section 4 lists properties of a successful RL method for this problem, which includes off-policy and "work with tree MDP instead of temporal MDP". Why is it important to learn off-policy? We know of many successful on-policy algorithms for RL, what am I missing here? Why do they have to work with tree MDPs?

The empirical setting is described in previous papers and the current paper relies on that. How is the training data generated? Do the problems differ in difficulty? Do we have to optimally solve the problem to attain the Markov property to then train the model? If so, how are the problems solved? Assuming that the training instances are easy (one needs to solve them optimally), how does the learned heuristic scale to larger problems?

The number of seeds also seems to be small (5), for the kind of learning being done.

Overall, it seems that the paper has some interesting ideas, but I don't fully understand them. The paper was written for people who already knows the details of this line of work, and it isn't friendly to newcomers to the point that the paper isn't self contained.

**Questions:**

I would like to hear clarifications on the empirical setup on how the training of the branching function is done, as I listed in the weaknesses section above.

---

> ### Author Response · Authors · 2023-11-21
> **Thank you for your review**
>
> > The main contribution of the paper, TreeDQN, is explained in a single paragraph in the main text.
>
> Our method is described in 3 pages (pp. 4 - 6). It includes analysis of distributions of tree sizes, proof of convergence of the tree Bellman operator, discussion of our proposed loss function, implementation details (use of Double Dueling DQN adapted for tree MDP process), and training details.
>
> > Overall the background section could be re-written to use less space and pack more information to help the reader understand the work.
>
> We updated the background section and improved the description of the Branch-and-Bound algorithm. Please see the updated version.
>
> > Why do we need to use DFS as node selection or set the global upper bound in the root to the optimal solution cost to guarantee the Markov property?
>
> We improved this section. Please see the updated version. To guarantee the Markov property, we need probabilities $p^+$ and $p^−$ depending only on the parent state and action. However, they depend on the global upper bound (GUB), which can vary for different visiting orders. To enforce the Markov property, one can either set the GUB in the root node equal to the optimal solution or choose Depth First Search as a node selection strategy.
>
> If GUB is set equal to the optimal solution, it will remain the same for every node of the tree since a node can not contain a solution better than the optimal.
>
> We also can compute GUB for the descendant nodes if we know the parent node and use a deterministic top-down visiting strategy like DFS.
>
> > The gap between training and testing is due to assuming that one has the optimal solution in training? Why not use DFS and not assume that you have the optimal solution in training?
>
> We do not assume that we have an optimal solution. In our experiments, we use DFS during training and switch to the SCIP default node selection strategy during testing since it is more efficient (but can violate the Markov property). In our experiments, we see that our TreeDQN agent performs close to the Imitation Learning agent and Strong Branching heuristic, so the gap between training and testing environments is sufficiently small.
>
> > How can more efficient heuristics for node selection also induce a gap between training and testing? And why is this important?
>
> More efficient heuristics may violate the Markov property ($p^+$ and $p^-$ depend only on the current state). If they do, the reinforcement learning method may not learn an optimal policy since the state would not contain all the information to choose an optimal action.
>
> It is important because we want to test and train our network on the same distribution. If we apply a different node selection strategy, it may lead to a different distribution of the number of descendant nodes, which could decrease the efficiency of our method. In our experiments, we do not see significant performance degradation.
>
> > Section 4 lists properties of a successful RL method for this problem, which includes off-policy and "work with tree MDP instead of temporal MDP". Why is it important to learn off-policy? We know of many successful on-policy algorithms for RL, what am I missing here? Why do they have to work with tree MDPs?
>
> We updated this section. Solving MILP tasks is time-consuming. We need sample efficient methods to learn a variable selection policy. Off-policy methods are generally much more sample-efficient than on-policy since they can reuse arbitrary old experiences during training. In the paper, we updated this requirement from "Be off-policy" to “Be sample efficient”.
>
> When solving a MILP task, the Branch-and-Bound solver splits the problem into subproblems and produces a binary tree. To train an RL agent, we need to map this tree to an episode. Considering the whole tree as a single episode under the tree MDP paradigm lets an agent directly optimize the size of the tree.
>
> > How is the training data generated?
>
> For each task we have a distribution of parameters and randomly sample task instances from that distribution.
>
> > Do the problems differ in difficulty?
>
> Yes. Depending on sampled parameters, the task could be easy (LP relaxation of the initial problem provides an integer feasible solution, so the size of the resulting B&B tree is 1) or require multiple branching decisions to find an optimal solution.
>
> > Do we have to optimally solve the problem to attain the Markov property to then train the model?
>
> No. In our work, we use DFS during training and the default node selection strategy during testing.

---

> ### Author Response · Authors · 2023-11-21
>
> > Assuming that the training instances are easy (one needs to solve them optimally), how does the learned heuristic scale to larger problems?
>
> In general, testing an ML model on out-of-domain tasks is hard. In our work, we demonstrate that our model can generalize to some extent to more complex tasks with a larger number of branching variables. The intuition of why it works is twofold:
> 1. The model learns a policy that can generalize to a larger number of branching variables.
> 2. During the solution, the domain of some variables is getting tightened to a single integer, and the actual amount of variables that can be branched decreases.
>
> > The number of seeds also seems to be small (5), for the kind of learning being done.
>
> We follow Gasse, Exact combinatorial optimization with graph convolutional neural networks., NeurIPS 2020 and Scavuzzo, Learning to branch with tree mdps, NeurIPS 2022 and use 5 random seeds during testing. Using a much larger amount of seeds would be difficult since solving MILP tasks is time-consuming.

---

### Author Response · Authors · 2023-11-21
**General response**

We thank all reviewers for their time and effort. Your comments helped us to improve our paper. Here is a brief summary of the changes:
1. We updated the description of the B&B algorithm.
2. We extended the explanation of the TreeMDP and made it self-contained.
3. We added SCIP with default parameters to our benchmarks. However, the internal branching rules can make modifications to the state of the solver and decrease the size of the resulting tree, so SCIP can not be considered a direct competitor to other methods. For more information refer to:
   1.  Discussion by Maxim Gasse (https://github.com/ds4dm/ecole/discussions/286#discussioncomment-2317487)
   2. Gerald Gamrath and Christoph Schubert. Measuring the impact of branching rules for mixed-integer programming. In Operations Research Proceedings 2017, pages 165–170. Springer, 2018.

4. We added additional test Balanced Item Placement from ML4CO competition (see Appendix D). The results in Appendix D demonstrate that our method outperforms Imitation Learning and Strong Branching in terms of total reward by a high margin. This result will not be possible with less sample-efficient methods since solving a single MILP instance of this problem requires 15 minutes.
5. We refined the text in general.

---

### Meta-Review · Area_Chair_ZKdB · 2023-12-09

**Metareview:**

The paper describes a new algorithm based on tree MDPs to improve the efficiency of MILP solvers by learning to branch.  The new algorithm yields smaller search trees and better run time than previous RL-based techniques.

As pointed out by the reviewers, the paper is difficult to read.  I also read the paper.  Depite the revisions made to the paper, it remains difficult to read.  I could not understand the notion of a tree MDP.  Tree MDPs are not a contribution of this paper, so one would expect a simple and clear description, but I had to read the Scavuzzo et al. to understand.  The main problem is that the paper never makes it clear that there are two next state distributions.  It looks like there is a single next state distribution with a probability of generating a left child and a probability of generating a right child.  This problem shows up again in Equation 2.  Equation 2 suggests that p+ and p- are probabilties that sum up to 1 since MDPs have a single next state distribution.  Something is missing in Equation 2.  There should be a sum over p+ with multiple next states  s+ and a sum over p- with multiple next states s- to reflect the fact that there are two expectations with respect to two next state distributions.  The other thing that is not clear is what are those proabilities p+ and p-.  The paper never describes them.  Scavuzzo et al. suggest that they are deterministic probabilities (i.e., 0 or 1), but it is not clear what are the next states and which next states have 0 vs 1 transition probability.  The paper explains that the states are (MILP_t and GUB_t), but MILP_t and GUB_t are not described explicitly.  The reward function is not defined either.  The paper says "if the reward is -1...", which suggests that some states have reward -1, but what are the other rewards and when does a state have a reward of -1 versus something else?  Finally, when does the process terminate?  I am assuming that the process terminates when an optimal solution or a contradiction is found, but it would be good to describe this and how backtracking gets reflected in the tree MDP.

The contraction in mean claimed in Theorem 4.1 is an important result to justify the proposed algorithm.  However, I'm not following the proof.  The key is to prove that p- + p+ < 1.  The paper claims that E(p- + p+) = n/n+1, but it is not clear how this is obtained. Since p+ and p- are supposed to be two distributions, how do we conclude that there sum is necessarily less than 1. Let's assume that E(p- + p+) is indeed n/n+1, then something is still missing in the proof.  For the contraction we need p- + p+ < 1 at each time step, not just in expectation.  Hence, it is not clear that the update operator of tree MDPs exhibit a contraction in mean.

As requested by the reviewers, additional results with the SCIP baseline were added to the paper.  This is good, but this raises some important questions.  The paper claims that the proposed algorithm improves the state of the art for RL-based techniques.  While this is the case, Table 3 shows that the non-RL techniques including SCIP generate smaller trees.  Some discussion is needed here.  What is it that allows non-RL based technique to generate smaller trees and why should someone care about RL-based techniques?

The paper indicates that the running time is proportional to the size of the trees, but the techniques with smaller search trees in Table 3 have longer run time in Table 4.  Some discussion is needed to explain this. It is good to see that the proposed technique achieves the best run time in general, but this seems to be due to something else than the size of the search tree.  Some explanation is needed.

**Justification For Why Not Higher Score:**

Lack of clarity regarding the problem definition, the analysis of the results and the proof of the main theorem.

**Justification For Why Not Lower Score:**

N/A

---

### Decision · Program_Chairs · 2024-01-16

Reject